# Effects of Vertical Smashing Rotary Tillage on Root Growth Characteristics and Yield of Broccoli

Fake Shan [1], Dongfang Li [1], Jianxi Zhu [2], Shuo Kang [1] and Jun Wang [1,*]

1   Department of Biosystems Engineering, Zhejiang University, 866 Yuhangtang Road, Hangzhou 310058, China; 21913001@zju.edu.cn (F.S.); 18331056491@163.com (D.L.); 12013009@zju.edu.cn (S.K.)
2   Zhejiang Agricultural Machinery Research Institute, 1158 Zhihe Road, Jinhua 321051, China; zhujianxi829@163.com
*   Correspondence: jwang@zju.edu.cn

**Abstract:** Most of the soils of the cultivated land in southern China are Ferralsols, which are easily deposited and hardened. To date, rotary tillage (RT) has been the major tillage system used in China. This tillage system results in a shallow soil pan, which reduces broccoli growth and yield. A two-year field experiment was conducted in the Central Zhejiang Basin, China, to compare the effects of vertical smashing rotary tillage (VSRT), RT, and vertical rotary tillage (VRT) on the soil properties, growth characteristics, and yield of broccoli. VSRT reduced the bulk density and penetration resistance of the 0–40 cm soil layer, and increased the soil water content of the 10–40 cm layer. Compared with RT and VRT, VSRT significantly promoted broccoli root length and increased broccoli root dry matter accumulation (DMA). VSRT significantly increased the DMA rate during the growth period, and the size of the broccoli florets was more uniform. In 2020, compared with RT and VRT, VSRT increased yields by 7.8% and 19.5%, respectively; while in 2021, the corresponding increases in yield due to VSRT were 24.8% and 40.5%. Therefore, VSRT, as a deep tillage method, can improve soil characteristics before planting broccoli and ultimately increase broccoli yield.

**Keywords:** ferralsols; vertical smashing rotary tillage; soil properties; broccoli growth characteristics; dry matter accumulation; yield





## 1. Introduction

Tillage is an important method that is used to change soil characteristics, improve soil physical and chemical properties, and create a good environment for crop growth [1–3]. Tillage usually means reducing soil hardness, improving soil structure, and improving soil water storage capacity through interactions between machines and the soil. Studies have shown that deep rotary tillage (RT) practices have a positive influence on crop growth [4]. Some studies have found that plowing before sowing results in marked effects on root growth and soil compaction (penetration resistance and soil bulk density), and may improve crop yield [5]. Studies have also reported that deeper tillage can alleviate subsoil compaction, thereby improving crop yield [6]. Apart from deep RT, conservation tillage, including less frequent tillage and no-tillage (NT), has also been the focus of research in recent years, and their effects on improving soil properties have been confirmed. However, recent studies have shown that less frequent tillage and no-tillage (NT) will affect soil properties, leading to shallow soil layer hardening and growth inhibition of crop roots, especially in deeper soil layers [7,8]. To address these problems, researchers have investigated the effects of different tillage practices. The results of these studies have shown that subsoiling can remarkably improve soil characteristics and physical properties, and increase the annual yield and economic performance of crops [9,10].

With the development of tillage machines, a wide variety of deep tillage methods have been applied to production. As a new type of deep tillage machine, vertical smashing rotary tillage (VSRT) machines are composed of five vertically installed spiral drills that can smash

the soil vertically. The theoretical maximum tillage soil depth of VSRT machines is 40 cm. To balance the working force, the helix direction and rotation direction of two adjacent cutters are opposite. Compared with common tillage methods, such as RT and VRT, the rotary smash tillage method has a deeper tillage depth, a better smashing effect, and looser soil after tillage, making it easier for root systems to access air and moisture [11]. At present, this type of tillage practice has been applied to some cash crops in China, including maize and potato [12–14]. Deep RT can increase crop yields, mainly by improving the crop root growth environment. Vegetables that have well-developed root systems require more space for root growth, which is why deep tillage is needed. However, in southern China, there is still a lack of research on the effects of VSRT on the growth process and yield of deep root vegetables.

Broccoli is a vegetable crop that is widely cultivated throughout the world, including in China [15]. Broccoli is rich in nutrition and is an important vegetable. Increasing the yield of broccoli can increase farmers' incomes. Compared with other vegetables, broccoli has a well-developed root system. The broccoli produced in Zhejiang Province, China, has compact floret balls, a slightly sweet taste, crisp stems, soft waxy florets, and a high quality. The main broccoli planting mode in the Central Zhejiang Basin is planting in autumn and harvesting in winter, with one crop grown each year. Before planting, deep soil loosening and high ridging are needed, to ensure that the broccoli root system has enough growth space. However, most of the cultivated soil types in the Central Zhejiang Basin are cohesive soil. Cohesive soil is easily deposited and compacted, and its penetration resistance is relatively high [16]. At present, RT is generally adopted before planting broccoli. Although RT crushes soil effectively, the tillage depth is shallow, usually less than 20 cm [5,17]. Therefore, under RT, the soil in deeper soil layers is not crushed. RT machinery is heavy, and this may cause secondary compaction of the deep soil layer. A solid soil pan is formed under long-term RT, which has a negative impact on root growth and nutrition acquisition, and further influences the yield of crops [18]. Therefore, according to different soil conditions and restrictive factors, choosing appropriate tillage practices is an important research direction, to combine agricultural machinery and agronomy and to meet the needs of agriculturally sustainable development [19,20]. This work mainly focuses on the effects of VSRT on (1) soil characteristics, including bulk density, penetration resistance, and water content; (2) dry matter accumulation during crop growth; and (3) crop yield.

## 2. Materials and Methods

### 2.1. Experiment Site

The field experiment was conducted at the Jinhua Agricultural Experimental Station, Zhejiang Province, China for two years, to study the effects of soil characteristics on broccoli floret yield and dry matter accumulation during the broccoli growth period in autumn and winter of each year. The experimental location was in the center of the Central Zhejiang Basin (29°01′ N, 119°48′ E). This area has a subtropical monsoon climate. The average temperature during the growing period is 17.33 °C, and the annual precipitation is about 1451.6 mm.

The soil type is classified as loamy clay. The experimental land is located on the cooperative experimental land of the Jinhua Academy of Agricultural Sciences, and no crops were planted for six months before the field experiment began. Randomized soil samples were collected from the 0.0–40.0 cm soil layer, homogenized together, air-dried, smashed, and screened (<2 mm), to determine the physical and chemical characteristics of the soil. In 2020, the soil chemical properties of the experimental field before the tillage treatment were as follows: pH 4.78, 14.2 g kg$^{-1}$ soil organic C, 0.9 g kg$^{-1}$ total N, 57.4 mg kg$^{-1}$ available *p*, 173 mg kg$^{-1}$ available K. In 2021, the soil physical and chemical properties of the experimental field before tillage treatment were as follows: pH 5.63, 18.6 g kg$^{-1}$ soil organic C, 1.1 g kg$^{-1}$ total N, 73.1 mg kg$^{-1}$ available *p*, 224 mg kg$^{-1}$ available K. There were excellent light, heat, and water conditions at the experimental station. The meteorological data for each month during the growing period are shown in

Table 1. The start time of the test was adjusted according to the climate of the year, and 2021 started earlier than 2020.

**Table 1.** Monthly meteorological data during the broccoli growth periods in 2020 and 2021.

| 2020 | | |
|---|---|---|
| **Month** | **Mean Temperature (°C)** | **Precipitation (mm)** |
| September | 24.13 | 172.9 |
| October | 19.42 | 16.4 |
| November | 16 | 27.4 |
| December | 7.86 | 21.9 |
| **2021** | | |
| August | 29.14 | 152.7 |
| September | 28.62 | 60.9 |
| October | 21.25 | 44.4 |
| November | 14.28 | 62.1 |

*2.2. Experiment Design*

"Youxiu" (Clover Seed Company Ltd., Hong Kong, China), an early-maturing broccoli variety that is widely planted in Zhejiang Province, was selected as the vegetable material for the experiment. Three tillage practice treatments, namely, RT, vertical rotary tillage (VRT), and VSRT, were used in the experiment. The machines used for the three different tillage methods are shown in Figure 1, and the specific tillage practice steps are described below. The tillage widths of RT, VRT, and VSRT were 1.8 m, 1.2 m, and 1.2 m. The designed tillage depths of RT, VRT, and VSRT were 20 cm, 25 cm, and 40 cm. The operation power of the three tillage machines was provided by a tractor (Kubota M954KQ, 70.8 kW). Except for tillage depth, there were differences between the VRT and VSRT machines. The spiral blades of the VSRT machine provided better soil smashing capacity and softer soil than the transverse blades of the VRT machine.

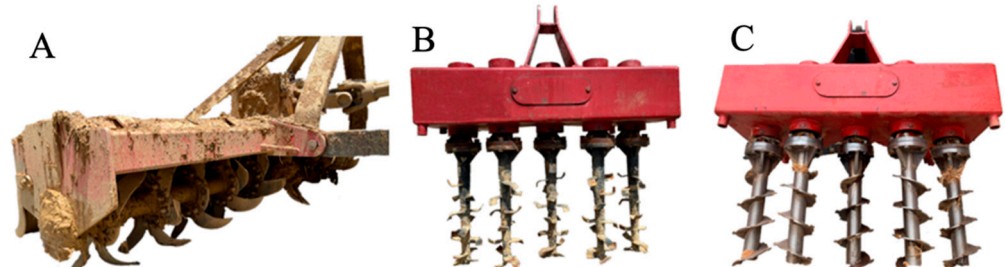

**Figure 1.** Machines used in the three tillage practices. (**A**): A rotary tillage machine under working conditions; (**B**): A vertical rotary tillage machine; (**C**): A vertical smashing rotary tillage machine.

RT: Fertilizer was spread on the soil surface using a fertilizer spreader → cultivated land was plowed twice with a RT machine to a depth of 20 cm → the ground was harrowed using a flat harrow linked to the RT machine.

VRT: Fertilizer was spread on the soil surface using a fertilizer spreader → cultivated land was plowed twice with a VRT machine to a depth of 25 cm → the ground was harrowed using a flat harrow linked to the RT machine.

VSRT: Fertilizer was spread on the soil surface using a fertilizer spreader → cultivated land was plowed twice with a VSRT machine to a depth of 40 cm → the ground was harrowed using a flat harrow linked to the RT machine.

After tillage, according to agronomic requirements, the land was left to stand for one week. Two different experimental plots were selected. In 2020, the experimental plot was 60 × 20 m, which was further divided into 15 experimental areas (12 × 6.6 m). In 2021, the experimental plot was 65 × 15 m, which was further divided into 15 experimental areas

(13 × 5 m). The plant spacing was 40 cm, and the row spacing was 60 cm. A randomized block design with five replicates was used to reduce the influence of the original soil fertility. The time nodes of the 2020 experiment were as follows: broccoli seeds were sown in the greenhouse using the hole plate seedling raising method on 25 September; seedlings were transplanted to the Jinhua experimental field on 10 October; and broccoli was harvested on 10 December. Due to the differences in breeding time, the time nodes of the 2021 experiment were as follows: broccoli seeds were sown in the greenhouse using the hole plate seedling raising method on 25 August; seedlings were transplanted to the Jinhua experimental field on 15 September; and broccoli was harvested on December 1.

The base fertilizer applied in the experiment was nitrate sulfur-based compound fertilizer produced by Garsoni Company, Chengdu, China (N-$P_2O_5$-$K_2O$: 15-5-26), and the topdressing was potassium sulfate type compound fertilizer produced by the Volcano Energy company, Shenzhen, China (N-$P_2O_5$-$K_2O$: 15-15-15). The base fertilizer was spread evenly on the ground before tillage and mixed into the soil by the different tillage methods. Topdressing refers to the spread of fertilizer on the soil surface between plants in the same row. In this study, topdressing was performed during the flowering period of broccoli. According to the growth characteristics of broccoli, field management measures were taken to control the negative effects of weeds, diseases, and pests on broccoli growth.

### 2.3. Sampling and Measurement

#### 2.3.1. Soil Water Content (SWC) and Soil Bulk Density (SBD)

A fixed volume (Φ50.46 × 50 mm, 100 $cm^3$) soil auger (Nanjing Soil Instrument Factory Co., Ltd., Nanjing, China) was used to collect soil samples, and samples were taken from the 0–40 cm soil layer. The samples were taken with a 10 cm interval depth. Five samples were taken from each experimental area before planting. The samples were placed into an oven and the SWC was determined using the oven-drying method. The equation for calculating SWC was as follows:

$$SWC(\%) = \frac{W_f - W_d}{W_d} \times 100$$

where $W_f$ and $W_d$ represent the fresh weight and dry weight of the soil, respectively.

The soil was transferred to a container, placed in an oven at 105 °C until a constant weight, and weighed. The bulk density was the oven-dry mass of the sample divided by the sample volume (100 $cm^3$).

#### 2.3.2. Soil Penetration Resistance

After tillage, soil penetration resistance was measured using a portable electronic soil penetration resistance tester (Nanjing Soil Instrument Factory Co., Ltd., Nanjing, China). Figure 2 shows the tester and the test process. The penetration resistance was measured in the range of the 0–40 cm soil layer, with an increment of 10 cm.

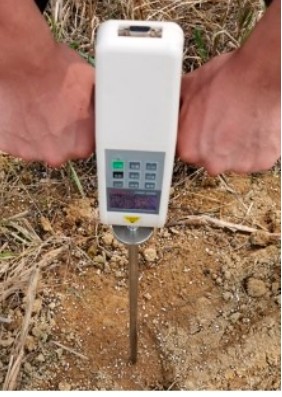

**Figure 2.** Portable electronic soil penetration resistance tester.

### 2.3.3. Dry Matter Accumulation

After eliminating broccoli with obvious defects, three uniform plants were randomly selected from each test plot as test samples. During sampling, the broccoli was cut at the demarcation point of the stem and root near the ground, and the whole plant was then split into the floret ball (during the floret ball growth period), stem, and leaves. To improve the drying efficiency and ensure drying quality, each part was cut into small pieces and then placed into an oven to measure dry matter accumulation (DMA). When measuring the DMA of the root, half of the plant distance (half of the distance between the base of two plants) between two plants was taken as the side length (40 × 60 cm), and the roots remaining in the soil were used as the center to create a square. The excavated soil samples were placed into nylon bags for root cleaning and picking, and the DMA of the sorted roots was measured using the oven-drying method.

### 2.3.4. Root Length

Five broccoli samples were taken from each experimental area to measure the root length. First, the midpoint of the diagonal was determined as the central sampling point, and then four points on the diagonal with the same distance from the central sampling point were selected. The distance from the position where the roots of the broccoli appeared to the longest root with a diameter greater than 1 mm was measured as the root length using a Vernier caliper.

### 2.3.5. Broccoli Characteristics and Yield

Broccoli characteristics and yield were measured at the same time as the root length. Characteristics including plant height, true leaf number, stem diameter, and broccoli floret ball diameter were measured throughout the broccoli growth period. The soil properties mentioned above were measured at the same time, to reveal relationships between broccoli growth characteristics and soil properties. The total yield under different tillage practices was measured at the harvesting stage.

### 2.4. Statistical Analysis

The experimental results were analyzed using analysis of variance (ANOVA) with SPSS software (ver. 24.0; SPSS Inc., Chicago, IL, USA). Graphs were plotted in OriginPro 2018C (OriginLab Corp., Northampton, MA, USA) and edited in Visio 2016 (Microsoft Corp., Redmond, WA, USA).

## 3. Results

### 3.1. Soil Water Content

Figure 3 shows the SWC of the 0–40 cm soil layer with a 10-cm soil sampling interval depth under the three different tillage methods. In 2020, except for the 0–10 cm soil layer, the SWC under VSRT was higher than that under RT and VRT, and a significant difference was observed in the 10–30 cm soil layer. The biggest difference was observed in the 20–30 cm soil layer; the SWC under VSRT was 1.6% and 1.9% higher than that with the RT and VRT, respectively. The opposite phenomenon was observed in the 0–10 cm soil layer, in which the SWC of VSRT was less than that of RT and VRT. In 2021, there was a significant difference in the SWC under VSRT in the 0–20 cm soil layer, and there was little difference at 30–40 cm. The biggest difference was observed in the 10–20 cm soil layer, the SWC under VSRT was 0.9% and 1.3% more than with the RT and VRT, respectively.

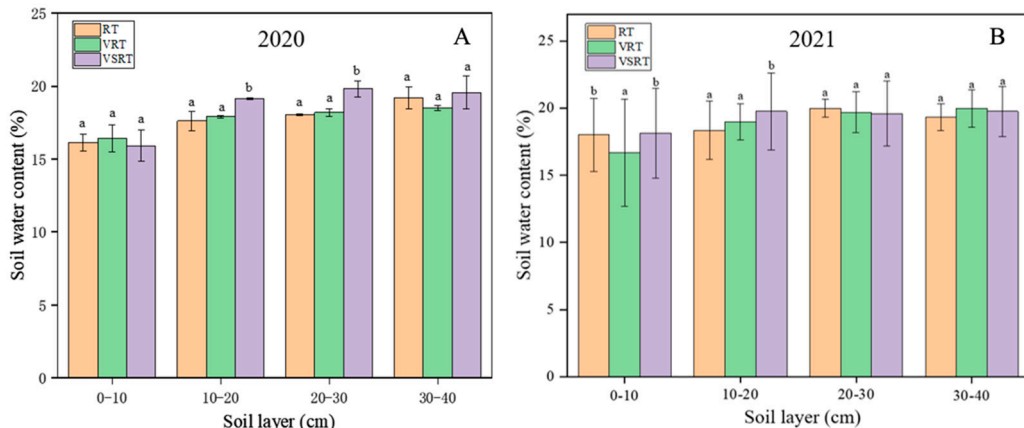

**Figure 3.** Soil water content of the 0–40 cm soil layer with the three tillage methods. (**A**): The soil water content in 2020; (**B**): The soil water content in 2021. Different lowercase letters in the same soil layer indicate significant differences at a $p < 0.05$ level. Rotary tillage, RT; vertical rotary tillage, VRT; vertical smashing rotary tillage, VSRT.

### 3.2. Soil Penetration Resistance

The soil penetration resistance of the three tillage practices is shown in Figure 4. Similar soil penetration resistance trends were observed in 2020 and 2021, although the maximum values differed. After tillage, VSRT significantly reduced the 10–40 cm soil penetration resistance compared with the other two methods, but the effect in the 0–10 cm soil layer was not remarkable. RT showed the highest soil penetration resistance among the three soil tillage practices. The biggest difference was observed in the 30–40 cm soil layer in 2020, the soil penetration resistance with VSRT was 52.6% and 41.8% less than with RT and VRT, respectively.

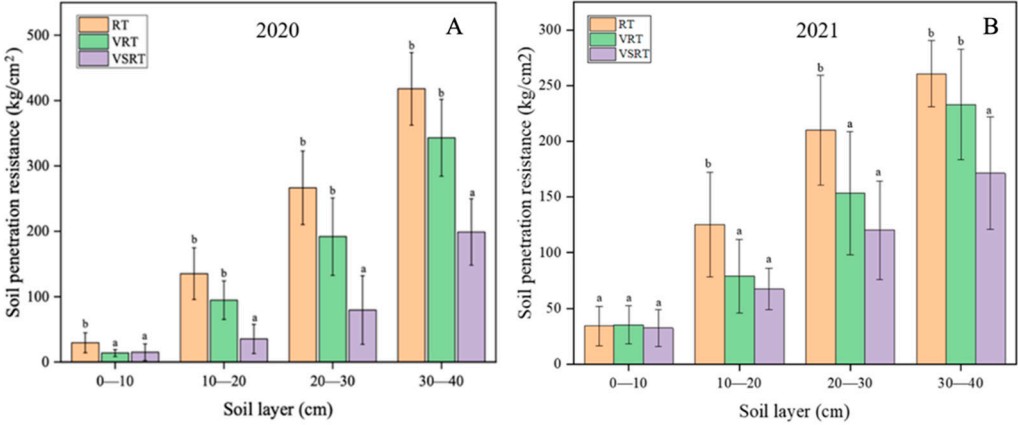

**Figure 4.** Soil penetration resistance in the 0–40 cm soil layer with the three tillage methods. (**A**): The soil penetration resistance in 2020; (**B**): The soil penetration resistance in 2021. Different lowercase letters in the same soil layer indicate significant differences at a $p < 0.05$ level. Rotary tillage, RT; vertical rotary tillage, VRT; vertical smashing rotary tillage, VSRT.

### 3.3. Soil Bulk Density in the 0–40 cm Soil Layer

Compared with RT and VRT, the SBD of the 0–40 cm soil layer decreased after VSRT treatment. There was a significant difference in the VSRT treatment in the 20–40 cm soil layer in 2020, but in the 0–20 cm soil layer, the difference was not significant (Figure 5). The largest difference was observed in the 20–30 cm soil layer, the SBD with VSRT was 7.4% and 8.8% less than with RT and VRT, respectively. In 2021, the SBD of the 10–40 cm soil layer showed a significant decline under VSRT. The biggest difference was observed in the 20–30 cm soil layer, the SBD under VSRT was 14.6% and 15.1% less than with RT and

VRT, respectively. In addition, the soil bulk density after VRT treatment was higher in the 0–30 cm soil layer compared with the others. There was little difference between the three tillage methods in the 0–10 cm soil layer.

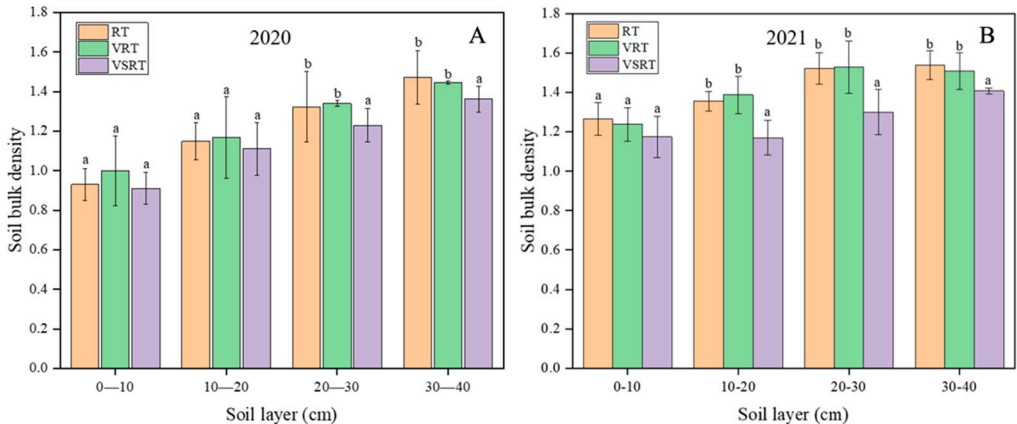

**Figure 5.** Soil bulk density of the 0—40 cm soil layer under three the tillage methods. (**A**): The soil bulk density in 2020; (**B**): The soil bulk density in 2021. Different lowercase letters in the same soil layer indicate significant differences at a $p < 0.05$ level. Rotary tillage, RT; vertical rotary tillage, VRT; vertical smashing rotary tillage, VSRT.

### 3.4. Broccoli Root Growth

As shown in Figure 6, in 2020, the root system of broccoli exhibited a rapid growth period within 30 days of planting, and the root length growth rates under the VSRT, RT, and VRT treatments reached the maximum within 10–20 days of planting. In a short time, the root system grew rapidly to about 20 cm. In 2021, the same rapid growth phenomenon was observed within 40 days of planting, and the maximum root length of broccoli under VSRT reached nearly 30 cm. In the two-year field experiment, compared with RT and VRT, VSRT resulted in a higher growth rate of the root length in 10–30 days. However, in 2020, from days 30 to 50, no significant difference was observed under the three practices, and the root length growth gradually slowed. Similar phenomena were observed from days 30 to 60 in 2021. In 2020, the root growth rate tended to be stable from days 50 to 70 during the growth period. A similar phenomenon was observed on days 60–80 in 2021. In the final harvest period of the two-year field experiment, the root length of broccoli under VSRT was significantly longer than that under RT and VRT, and the change in root DMA was similar to that of root length. Compared with RT and VRT, the final root DMA under VSRT increased significantly. In general, throughout the growth period, the root system growth of broccoli under the VSRT treatment was more advantageous.

### 3.5. Characteristics of Broccoli

The characteristics data of broccoli during the final harvest period are listed in Table 2. In both years, under VSRT, broccoli floret balls were slightly wider, had diameters that fluctuated within a smaller range, and were more uniform in size, which made the florets more valuable commodities. In addition, the stem diameter of the broccoli under VSRT was significantly different from that of RT and VRT in 2020, and the stem diameter under VRT was significantly different from that of RT and VSRT in 2021. The highest plant height was measured under the VSRT treatment in 2020, and the highest plant height was measured under RT in 2021.

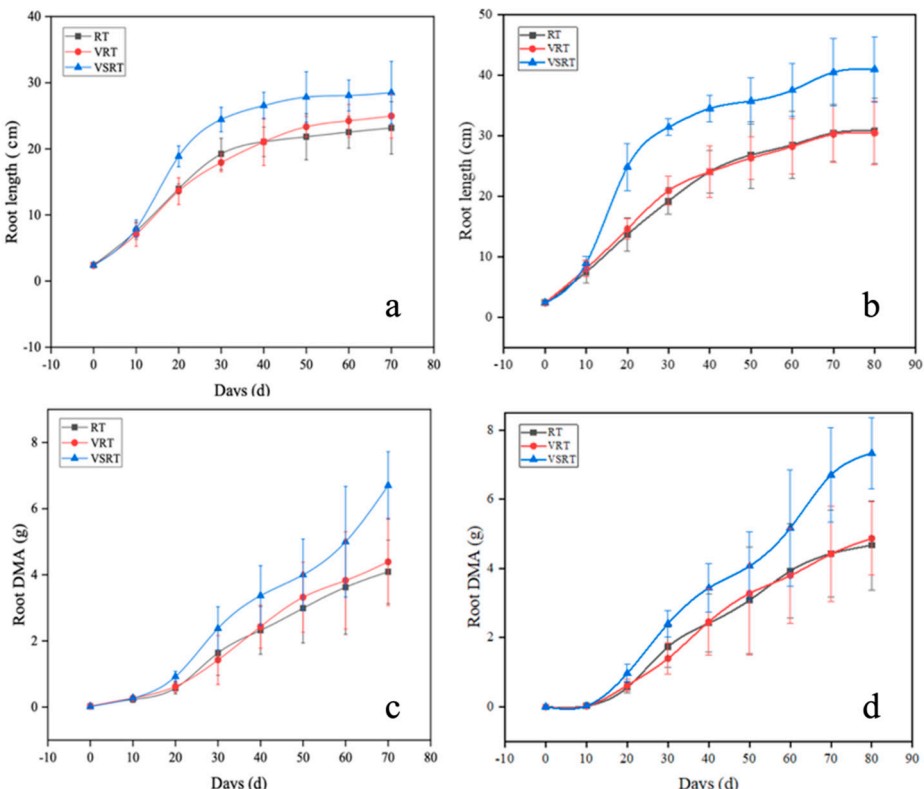

**Figure 6.** Dynamic changes of root elongation and dry matter accumulation (DMA) of broccoli under the three tillage methods in 2020 (**a**,**c**) and 2021 (**b**,**d**). Rotary tillage, RT; vertical rotary tillage, VRT; vertical smashing rotary tillage, VSRT.

**Table 2.** Characteristics of broccoli under the three tillage practices in 2020 and 2021.

| Tillage Practices | Plant Height (cm) | Stem Diameter (cm) | Floret Ball Diameter (cm) |
|---|---|---|---|
| **2020** | | | |
| RT | 51.4 ± 4.7 a | 1.93 ± 0.11 a | 14.25 ± 0.75 a |
| VRT | 53.2 ± 2.3 a | 1.86 ± 0.08 a | 13.94 ± 0.84 a |
| VSRT | 58.3 ± 3.8 b | 2.33 ± 0.09 b | 14.56 ± 0.33 a |
| **2021** | | | |
| RT | 57.5 ± 6.7 a | 3.54 ± 0.23 a | 16.76 ± 0.26 a |
| VRT | 52.3 ± 4.3 b | 3.28 ± 0.13 b | 14.96 ± 1.75 b |
| VSRT | 56.2 ± 5.3 a | 3.52 ± 0.17 a | 18.58 ± 0.38 a |

Different lowercase letters in the same soil layer indicate significant differences at a $p < 0.05$ level. Rotary tillage, RT; vertical rotary tillage, VRT; vertical smashing rotary tillage, VSRT.

*3.6. DMA*

Figure 7 shows the DMA of broccoli plants and floret balls, as well as the DMA rates. In 2020, the DMA rate of the broccoli plants under VSRT reached a peak 20–30 days after planting, and the DMA was significantly different from the RT and VRT values at the same time point. At 30–60 days, the plant growth rate decreased and at 60–70 days it increased. Finally, the plant DMA under VSRT was always greater than that under the RT and VRT; but in 2021, during the last growth period, the plant DMA under RT exceeded that of VSRT and VRT. The growth rate of broccoli floret balls was relatively stable, and the DMA of floret balls under VSRT was significantly different on days 20–30 in both years. In 2020, there was little difference in the DMA of floret balls at harvest under the different tillage methods, and the order of DMA from greatest to least was VSRT > RT > VRT. In 2021, the DMA of the floret balls varied greatly under the different tillage methods, and the order of DMA from greatest to least was VSRT > RT > VRT.

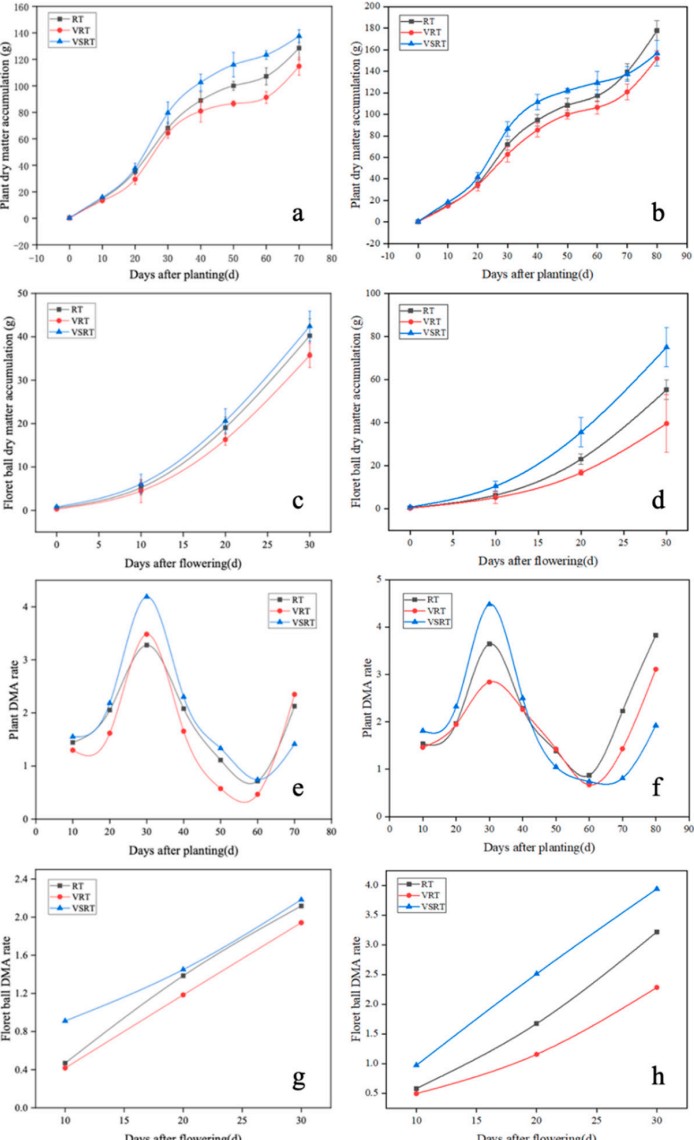

**Figure 7.** Dry matter accumulation (DMA) of broccoli plants and floret balls, and the DMA rate of broccoli plants and floret balls. (**a**,**c**,**e**,**g**) show values in 2020, and (**b**,**d**,**f**,**h**) show values in 2021. Rotary tillage, RT; vertical rotary tillage, VRT; vertical smashing rotary tillage, VSRT.

### 3.7. Broccoli Floret Ball Yield

Under the three tillage practices, significant differences in broccoli yield were observed (Figure 8). The broccoli yield of VSRT was higher than that of VRT and RT, and exhibited a significant difference compared with VRT. In 2020, compared with RT and VRT, VSRT increased the floret ball yield by 7.8% and 19.5%, respectively. In 2021, compared with RT and VRT, VSRT increased the floret ball yield by 24.8% and 40.5%, respectively.

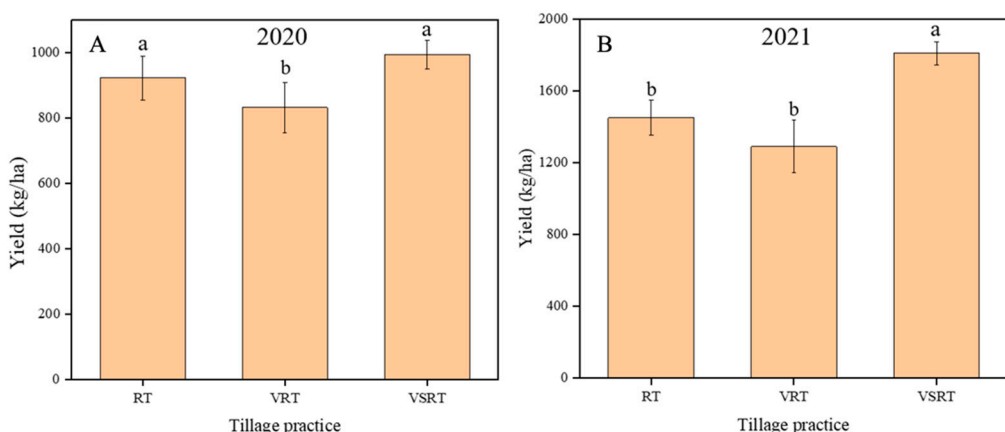

**Figure 8.** Broccoli yield under the three tillage practices. (**A**): The broccoli yield in 2020; (**B**): The broccoli yield in 2021. Different lowercase letters in the same soil layer indicate significant differences at a $p < 0.05$ level. Rotary tillage, RT; vertical rotary tillage, VRT; vertical smashing rotary tillage, VSRT.

## 4. Discussion

### 4.1. Effect of VSRT on Soil Properties

Other than no tillage, low tillage, and other conservation tillage treatments, the main soil management methods used in China are shallow soil layer tillage methods, such as RT. The use of RT has many advantages, such as a high efficiency, stable depth, and loose soil; but in most cases, RT cannot reach deeper tillage depths [16]. When RT is used for long periods of time, only the shallow soil layer improves. The weight during RT implementation is great, and the repeated use of RT causes compaction in the deep soil layer, which increases each time tillage is performed [17]. Ultimately, deep soil that is missed by the RT method for long periods of time will harden. The SBD and penetration resistance of this deep soil layer increase, forming a solid soil pan [16]. In most regions of China where RT is used, the depth of the plough soil layer moves up, to varying degrees [5]. Adverse effects have been observed on broccoli root growth in the hard soil pan soil layer [21]. Tillage practices are the most important agricultural methods used to change soil properties and the growth of plant roots [22]. Figure 4 shows that, compared with RT and VRT, the soil penetration resistance under VSRT was reduced. A significant difference between tillage treatments was observed in the 10–40 cm soil layer. VSRT also reduced the SBD, especially in the 20–40 cm soil layer. The SBD under VRT was greater than under RT. VRT has a high-speed rotating transverse blade, so the soil after cultivation has a high degree of fragmentation. The highly crushed soil is easy to deposit naturally with the influence of rain, and the SBD is high after deposition. Thus, while RT and VRT also improve the shallow soil layer, VSRT can improve the soil properties of deeper soil layers [12]. According to the different structures of the machines, the reasons for this are as follows: First, the spiral structure of VSRT leads to its good penetration and crushing effect on the soil pan. In addition, the structure of the machine spiral blade has a good lifting effect on the soil, so the soil after tillage is loosened, and the SBD and penetration resistance are greatly reduced.

SWC is an important factor affecting broccoli growth [23], and it significantly promotes root growth [24]. Compared with the other practices, VSRT increased the SWC, especially in the 10–30 cm soil layer. The reason for this effect may be that VSRT increases the soil looseness and water permeability, thus increasing the rainfall interception and infiltration speed, further improving the depth distribution of soil moisture and increasing the water storage capacity of soil [25,26]. A previous study found that deep tillage practices can expand the effective water pool available for roots and increase the scope of available water resources for roots [9]. Deep tillage can increase the abundance of mesopores in the soil, which expands the space available for soil water storage [9,27]. In addition, deep tillage can increase the infiltration rate of surface water, reduce surface runoff, and increase soil water storage [28]. Compared with conventional RT, VSRT significantly improved soil

properties in the 20–40 cm soil layer. Therefore, in long-term RT cultivated land, the deeper soil layer needs to be broken by VSRT. In addition, except for tillage effects, changes in soil properties are closely linked to climate. Increased rainfall leads to increased SWC and decreased soil penetration resistance. Wet soil is more easily compacted, thus leading to an increase in SBD. Therefore, in areas with high rainfall, VSRT can improve water use efficiency. There were some differences in soil characteristics between 2020 and 2021. The analysis of meteorological information suggests that these differences were mainly caused by changes in precipitation. Compared with 2020, there was more rainfall each month, resulting in higher SWC and lower soil penetration resistance in 2021, in the 0–20 cm soil layer. However, as long-term wet soil leads to denser soil, there was higher SBD in 2021.

*4.2. Effect of VSRT on Broccoli Root Growth*

Compared with RT and VRT, VSRT can mix deeper soil with shallow soil, loosen the soil layer after tillage, and increase the number of mesopores in the soil, which can increase SWC and reduce root growth resistance, to provide a more suitable environment for the growth of broccoli roots [29,30].

As shown in Figure 6a, compared with RT and VRT, the broccoli root length under VSRT had a rapid elongation period during days 10–20 of the growth period. This may have been because the growth range of broccoli roots was concentrated in the 10–20 cm layer during that period. Combined with the results shown in Figure 3, VSRT significantly increased the SWC in the 10–20 cm soil layer, reduced the soil penetration resistance and SBD in the 10–20 cm soil layer, and provided a more suitable environment with more loose and porous soil, deeper tilth layer, more SWC, and a more uniform distribution of soil nutrients, which increased the biomass of the root system [31–33]. During days 50–70, the elongation of the broccoli roots was not significant, but the dry matter weight of broccoli roots continued to increase. This may have been because, after the broccoli axial roots grew to a certain length, elongation in the vertical direction was reduced, and the growth of the broccoli root systems shifted to increase the number and length of lateral roots. VRT had higher soil fragmentation and reduced the resistance of broccoli root growth, so the root length of broccoli was longer than with RT.

*4.3. Effect of VSRT on Broccoli Yield*

The use of suitable tillage methods can improve vegetable yields. Different tillage methods mainly affect broccoli yield by changing the physical and chemical properties of the soil. In this study, by comparing the changes of DMA in different parts of broccoli during the growth process, differences were observed in the DMA of broccoli plants and floret balls under different tillage methods. Broccoli plants exhibited a rapid growth period under VSRT treatment, especially during the first 20–30 days of the growth process. This phenomenon may have been due to the large water demand in the early growth stage of broccoli [21]. VSRT treatment increased the soil water storage capacity and SWC, which provided sufficient water for broccoli growth. The growth characteristics of broccoli plants are affected by water stress, and tillage practices can affect SWC [23]. In addition, some studies have reported that salt stress in the soil is a factor that significantly affects the growth of broccoli, and VSRT can reduce soil salinity [34–36]. In 2021, although the soil penetration resistance was lower than that of 2020 among the three methods, combining the results of SWC, soil penetration resistance and SBD, the soli was soft but not loose, and the porosity was very low. This phenomenon further affected the root respiration and crop growth.

Compared with RT and VRT, VSRT increased the yield of broccoli. This effect was mainly because VSRT improved several soil properties affecting the growth of broccoli, such as the soil porosity, SWC, and soil hardness [15,37]. Compared with VRT, RT increased the yield of broccoli. This effect was mainly because VRT had a higher soil fragmentation and higher SBD, which inhibited the broccoli root respiration, thus reducing the broccoli DMA. The soil penetration resistance under RT was high, but the overall soil penetration

resistance was low, the effect of soil penetration resistance on broccoli growth was smaller than that of SBD and SWC. In 2021, the increase was more significant. This was mainly due to increased precipitation; the experimental field soil was wet throughout the growth period in 2021. The soil under the VSRT treatment was looser and more porous, which was more conducive to the crop growth. However, in 2020, the late growth period of broccoli was affected by diseases that impeded the growth of the flower balls, resulting in a decrease in broccoli yield. Therefore, VSRT, as a new deep tillage method, can be used to improve the adverse effects of long-term RT on soil properties and to increase broccoli yields.

## 5. Conclusions

VSRT treatment improved the soil properties, reduced SBD and penetration resistance, increased SWC, and provided a more suitable environment for broccoli growth. Under VSRT tillage, the broccoli DMA rate and root growth rate increased. In addition, VSRT improved broccoli the yield through improved broccoli floret ball weight. Therefore, the present research results can provide some guidance for the application of new farming methods and tools for improving the quality of cultivated land in southern China.

**Author Contributions:** Conceptualization, F.S. and D.L.; methodology, F.S.; software, F.S.; investigation, F.S.; data curation, S.K.; writing—original draft preparation, F.S.; writing—review and editing, J.W.; visualization, F.S.; supervision, J.W.; project administration, J.W.; funding acquisition, J.Z. All authors have read and agreed to the published version of the manuscript.

**Funding:** The authors would like to acknowledge the financial support of the Zhejiang SanNongLiu-Fang agricultural science and technology cooperation project in China through project 2020SNLF006.

**Institutional Review Board Statement:** Not applicable.

**Data Availability Statement:** Not applicable.

**Conflicts of Interest:** The authors declare no conflict of interest.

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
