# Peer review of "Effects of Vertical Smashing Rotary Tillage on Root Growth Characteristics and Yield of Broccoli"

_agriculture, doi:10.3390/agriculture12070928_

Round 1

Reviewer 1 Report

Effects of vertical smashing rotary tillage on the root growth characteristics and yield of broccoli

Review Report

This paper has a very good subject as it studied the effect of a new type of tillage on broccoli production, beside the effect of soil's physical properties on production, there were appreciative efforts have been made to serve the objectives of scientific research in this paper over two years, but there are many weaknesses need to fix.

Introduction

  • For scientific necessity and because it is an important technique and become used world widely you should write a brief about zero and minimum tillage advantages and disadvantages.
  • Write briefly about the advantage of the conservation tillage.
  • Wright brief about the effect of tillage in general on soil physical properties.
  • Some researchers have published work showing that conservation tillage increased crop production over conventional tillage, and other researchers found the opposite that conventional tillage increase crop production over conservation tillage. Please conclude the literature review for both teams.
  • Readers and researchers want to know why you want to increase broccoli production. What is the importance of broccoli and what is the benefit of increasing its production? Write a paragraph to show why it is needed to increase broccoli production, and what the importance of broccoli is economically and as a food.
  • Please write briefly about and differentiate between Rotary Tillage (RT), Deep Vertical Rotary Tillage (DVRT), Vertical Smashing Rotary Tillage (VSRT), and Smash Ridging Tillage (SRT).
  • Lines 31 and 32: (Deep rotary tillage is an important way to change soil characters, improve soil physical and chemical properties, and create a good environment for crop growth [1–3]) reference [2] showed that conservation tillage (M: Mulching; F: Ridge and Furrow Planting without Mulching; F–M: Ridge and Furrow Planting with Mulching) and NT non-tillage also improved the yield of wheat and maize, not just the deep tillage, please revise.
  • Lines 37-39: (In addition, less tillage will affect soil properties, leading to shallow soil layer hardening and inhibiting the growth of crop roots, especially in deeper soil layers [2,7,8].) But reference [8] concludes in its abstract: (The results indicated that RT increased the spatial and temporal root distribution, enhanced photosynthetic activities at the flowering stage, and achieved higher average grain yield by 12.0 % and 6.7 % from 2008 to 2019 as compared with NT and CT, respectively) and (Thus, RT is an effective strategy to improve grain yield and WUE of winter wheat in the NCP. In the future, the strategic tillage based on NT and RT may be a promising approach to sustain the benefits of NT (e.g., soil quality and water consumption) and RT (e.g., grain yield and WUE). NT = no-tillage, and RT = rotary tillage. Similarly, for reference [2] see the previous Paragraph. So what is concluded here is not right. Please revise.
  • Lines 48-51: Is the tillage machine and tool that was studied in reference [11] (smash-ridging machine) the same as VSRT?
  • Lines 51 and 52: (Up to present, this kind of tillage practice has been applied to some cash crops in China, including peanut, rice and maize [12–14].). These references included studies about summer maize, Potato, and Summer Maize, respectively not as you mentioned, please revise
  • Lines 62 - 64: )Before planting, deep soil loosening and high ridging are needed to ensure that broccoli root system has enough growth space [16](. This reference [16] was about maize, not broccoli, please revise.
  • Lines 64 - 66: However, most of the cultivated soil types in the central Zhejiang basin are cohesive soil, which is easy depositing and compacting, and the penetration resistance is relatively high [17]. This reference [17] about rotary tillage in Malaysia, please revise.
  • Lines 71 - 73: (Solid plough bottom is formed under a long-term maintenance of rotary tillage, which makes a negative impact on root growth and nutrition acquisition, further influences the yield of crops [3,19–23]). Reference [3] about crop rotation and did not mention rotary tillage, [19] studied Soil compaction–N interactions in barley, and the plants were grown in plastic cylinders no mentioning rotary tillage, reference [20] mentioned (Increasing size of agricultural implements……etc.) not just rotary tillage, reference [21]: (The repeated plowing in conventional tillage commonly results in the formation of dense plow pans that may seriously affect root growth) not rotary, also revise reference [22]. Just reference [23] including what you mentioned above. Please revise your references. Also, you need to edit your English writing ([to deposit] instead of: [depositing[; [, and further influencing] instead of: [, further influences] 
  • Lines 75 - 76: (In order to achieve the goal of combining agricultural machinery and agronomy, and at the same time, to meet the needs of agricultural sustainable development [25]). It seems that something is missing here, or maybe this is a completion of the previous sentences? If so, why do you separate the two sentences? Please revise.

Methodology

  • Show the technical specification of the tillage machines and tools. What is the width of tillage tools cm?
  • Ignoring the working depth, show the technical differences between VRT and VSRT machines.
  • Why you did not follow the same time nodes for both years? This is is to avoid the effect of the different climates, which you confirmed in lines 359 to 361: (On the other hand, in 2020, the late growth period of broccoli was affected by diseases, which impeded the growth of flower balls, 360 resulting in the decline of broccoli yield).
  • Line 121-122: (In 2020, the experiment plot was 60 × 20m which is divided into 15 experimental areas) what is the area for each plot? Is it 20 m × 4 m? And the same question for Line 122-123: (In 2021, the experiment plot was 65 × 15m which is divided into 15 experimental areas). Is it 20 m × 4.33 m? Show the area of each plot.
  • You should show the diameter and the height of the auger cylinder that was used for collecting bulk density data.
  • You should show how many samples you took per plot for bulk density and soil penetration resistance, and when did you take the samples before and after tillage, because you have two tillage in every experiment (Plough cultivated land two times with rotary tillage machine → Harrowing ground using the flat harrow linked to rotary tillage machine). So show when did you collect the data? Before rotary and after harrowing? Or immediately before planting?
  • Show clearly that you collected soil samples for bulk density and water content with a 10 cm soil sampling interval depth.
  • You have collected the data before and after plowing, it would be very clear to show the effect of tillage on soil physical properties by comparing physical properties before and after tillage.
  • How did you determine the soil moisture content? You should show the procedure and the equation.
  • Your calculation method of the bulk density is confusing, what do you mean by the weight of the soil auger? Did you put the soil auger in the oven? What do you mean by fresh soil? Is it the soil before drying? I'm confused I do not know what you mean. You should weigh the soil sample before putting it in the oven and after that. Use the dry soil mass to calculate the bulk density, and the weight difference between wet and dry soil to calculate the moisture content. This is not written here. I do not know from where you got the equation that you used to calculate the bulk density. The bulk density is calculated as (oven-dried soil mass/soil volume). You mentioned reference (26), but this reference did not mention this equation. This reference (Methods of Soil Analysis: Part 1 Physical and Mineralogical Methods, 5.1, Second Edition) stipulated on page 366 that: (Transfer the soil to a container, place it in an oven at 105°C until constant weight is reached, and weigh it. The bulk density is the oven-dry mass of the sample divided by the sample volume). You need to make a major explanation and revision here.
  • Lines 158 and 159: (After tillage practice, the soil penetration resistance was measured by the soil penetration resistance tester) is this tester a soil penetrometer? If so, what is the cone angle of the penetrometer (o) and the conical point (cm2) of the penetrometer that you used?
  • Show a picture of the soil penetration resistance tester (Nanjing Soil Instrument Factory Co., Ltd).
  • Line 174-176: Root length, how many samples per plot did you take, and how did you choose it?
  • Line 178-179: Broccoli characters & yield. How many samples per plot did you take and how did you choose them?
  •  
  1. Result
  • In your figures (figure 5 and figure 6) put every two similar factors together in a horizontal line, this would make it easy to compare.
  • Do you believe that deep plowing will always contribute to an increase in the broccoli yield, noting the comparison between RT and VRT, and do you see that your results are definitive and decisive? Maybe the deeper is most effective.
  • Please show the differences % (the increment or decrement ratio) between each soil property before and after tillage for RT, VRT, and VSRT. So readers could see the effect ratio.
  1. Discussion
  • Although VRT is deeper and the tillage tool is lighter than RT, your result showed that RT in the top layer of soil has a lower bulk density than VRT in 10 to 30 cm as in Figure 4, please explain and justify.
  • What was the effect of root length and DMA on broccoli yield, with notice that root length and DMA under VRT were mostly higher than that under RT? And the yield under RT was higher than under VRT. You should explain and discuss this.
  • The Soil penetration resistance under RT was higher than that under VRT in all soil layers during both years, and plant DMA, DMA accumulation, and DMA rate of Broccoli plant floret ball, florets ball diameter (cm), and the broccoli yield under RT were higher than that under VRT, is the effect of penetration resistance is ineffective or what? Discuss and explain.
  • Lines 297 and 298 (so the soil after tillage is looser), the right is to say (so the soil after tillage is loosened).
  • Lines 316 and 317: (Compared with 2020, there is more rainfall in each month, resulting in higher SWC and lower SPR in 2021. However, long-term wet soil leads to more dense soil, so there is higher SBD in 2021). You mentioned that bulk density in 2021 was higher but in line 95 you mentioned bulk density in 2020 was 1.47 g/cm3 and in line 98 you mentioned bulk density in 2021 was 1.42 g/cm3, explain. Is the effect of higher water content on bulk density positive or negative? What is the relation between soil penetration resistance, bulk density, and soil water content?
  • Lines 316 and 317: (Compared with 2020, there is more rainfall in each month, resulting in higher SWC and lower SPR in 2021). But when we look at Figure 2 we see that the SWC in 2020 in the 10 to 40 cm layer was higher than that of 2012, explain.
  • Lines 361 to 363 (Therefore, the results of this study show that VSRT, as a new deep tillage method, can be used to improve the adverse effects of long-term rotary tillage on soil properties and increase the yield of broccoli [44,45]). When returning to reference [44] which is: (de Pascale, S.; Maggio, A.; Barbieri, G. Soil Salinization Affects Growth, Yield and Mineral Composition of Cauliflower and Broccoli. European Journal of Agronomy 2005, 23, 254–264, doi:10.1016/j.eja.2004.11.007), and reference [45] which is (Wurr, D.C.E.; Hambidge, A.J.; Fellows, J.R.; Lynn, J.R.; Pink, D.A.C. The Influence of Water Stress during Crop Growth on the Postharvest Quality of Broccoli. Postharvest Biology and Technology 2002, 25, 193–198, doi:https://doi.org/10.1016/S0925-5214 (01) 00171-5) there is no any mentioning of VSRT, the first paper about planting stress (water and heat stress) and the second one about soil salinity. I do not know from where you got that. When you cite a reference you should be sure that what you referred to this reference is exist in it.
  • You should be sure about your references that they include the information you referred to them, please check all your references.

Author Response

Response to Reviewers' comments

>Reviewer #1:

This paper has a very good subject as it studied the effect of a new type of tillage on broccoli production, beside the effect of soil's physical properties on production, there were appreciative efforts have been made to serve the objectives of scientific research in this paper over two years, but there are many weaknesses need to fix.

Introduction

>1) For scientific necessity and because it is an important technique and become used world widely you should write a brief about zero and minimum tillage advantages and disadvantages.

>2) Write briefly about the advantage of the conservation tillage.

Response: I added a brief introduction about conservation tillage, including less frequent tillage and no-tillage, advantages and disadvantages, and the explanation was as follows (line 36-39):

Except deep RT, conservation tillage, including less frequent tillage and no-tillage (NT), were also the focus of research in recent years, and their effects on improving soil properties has been confirmed, but recent studies have shown that less frequent tillage and no-tillage (NT) will affect soil properties, leading to shallow soil layer hardening and inhibiting the growth of crop roots, especially in deeper soil layers [7,8].

>3) Wright brief about the effect of tillage in general on soil physical properties.

Response: I added a brief introduction about the effect of tillage in general on soil physical properties, and the revises were as follows (line 29-31):

Tillage usually means to reduce soil hardness, improve soil structure and improve soil water storage capacity through the interaction between machines and soil.

>4) Some researchers have published work showing that conservation tillage increased crop production over conventional tillage, and other researchers found the opposite that conventional tillage increase crop production over conservation tillage. Please conclude the literature review for both teams.

Response: According to your suggestions, I looked over the paper and checked the references carefully, and modified confusing references. Please see below for specific modifications.

>5) Readers and researchers want to know why you want to increase broccoli production. What is the importance of broccoli and what is the benefit of increasing its production? Write a paragraph to show why it is needed to increase broccoli production, and what the importance of broccoli is economically and as a food.

Response: I wrote a paragraph to show why it is needed to increase broccoli production, and what the importance of broccoli is economically and as a food. The explanation was as follows (line 60-61):

Broccoli is rich in nutrition and is an important vegetable. It is widely planted in China. Increasing the yield of broccoli can increase the income of farmers.

>6) Please write briefly about and differentiate between Rotary Tillage (RT), Deep Vertical Rotary Tillage (DVRT), Vertical Smashing Rotary Tillage (VSRT), and Smash Ridging Tillage (SRT).

Response: The difference between RT, VRT and VSRT was the unique structure of VSRT machine. SRT and VSRT had similar operation modes. The difference between SRT and VSRT was the structure of the spiral blade of the machine. The explanation was as follows (line 45-53):

With the development of tillage machines, a wide variety of deep tillage methods have been applied to production practice. As a new type of deep tillage machine, vertical smashing rotary tillage (VSRT) machines are composed of five vertically installed spiral drills, which can smash the soil vertically. The theoretical maximum tillage soil depth of VSRT machines is 40 cm. In order to balance the working force, the helix direction and rotation direction of two adjacent cutters are opposite. Compared with common tillage methods such as RT and VRT, this rotary smash tillage method has deeper tillage depth, better smashing effect, and looser soil after tillage, making it easier for root systems access to air and moisture [11].

>7) Lines 31 and 32: (Deep rotary tillage is an important way to change soil characters, improve soil physical and chemical properties, and create a good environment for crop growth [1–3]) reference [2] showed that conservation tillage (M: Mulching; F: Ridge and Furrow Planting without Mulching; F–M: Ridge and Furrow Planting with Mulching) and NT non-tillage also improved the yield of wheat and maize, not just the deep tillage, please revise.

Response: I checked the references carefully, modified inaccurate descriptions, and the revises was as follows (line 28-32):

Tillage is an important method used to change soil characteristics, improve soil physical and chemical properties, and create a good environment for crop growth [1–3]. Studies have shown that deep rotary tillage (RT) practices have a positive influence on crop growth [4].

>8) Lines 37-39: (In addition, less tillage will affect soil properties, leading to shallow soil layer hardening and inhibiting the growth of crop roots, especially in deeper soil layers [2,7,8].) But reference [8] concludes in its abstract: (The results indicated that RT increased the spatial and temporal root distribution, enhanced photosynthetic activities at the flowering stage, and achieved higher average grain yield by 12.0 % and 6.7 % from 2008 to 2019 as compared with NT and CT, respectively) and (Thus, RT is an effective strategy to improve grain yield and WUE of winter wheat in the NCP. In the future, the strategic tillage based on NT and RT may be a promising approach to sustain the benefits of NT (e.g., soil quality and water consumption) and RT (e.g., grain yield and WUE). NT = no-tillage, and RT = rotary tillage. Similarly, for reference [2] see the previous Paragraph. So what is concluded here is not right. Please revise.

Response: I looked over the paper carefully, modified inaccurate descriptions and reference [2]. The revises was as follows (line 36-41):

Except deep RT, conservation tillage, including less frequent tillage and no-tillage (NT), were also the focus of research in recent years, and their effects on improving soil properties has been confirmed, but recent studies have shown that less frequent tillage and no-tillage (NT) will affect soil properties, leading to shallow soil layer hardening and inhibiting the growth of crop roots, especially in deeper soil layers [7,8].

>9) Lines 48-51: Is the tillage machine and tool that was studied in reference [11] (smash-ridging machine) the same as VSRT?

Response: The smash-ridging machine is not the same as VSRT, but the tillage principle is similar. I looked over the paper carefully, and modified inaccurate descriptions. The explanation was as follows (line 50-53):

Compared with common tillage methods, this rotary smash tillage method has deeper tillage depth, a better smashing effect, and looser soil after tillage, making it easier for root systems access to air and moisture [11].

>10) Lines 51 and 52: (Up to present, this kind of tillage practice has been applied to some cash crops in China, including peanut, rice and maize [12–14].). These references included studies about summer maize, Potato, and Summer Maize, respectively not as you mentioned, please revise

Response: I looked over the reference carefully, and modified inaccurate descriptions. The revises was as follows (line 53-54):

At present, this type of tillage practice has been applied to some cash crops in China, including maize and potato [12–14].

>11) Lines 62 - 64: )Before planting, deep soil loosening and high ridging are needed to ensure that broccoli root system has enough growth space [16](. This reference [16] was about maize, not broccoli, please revise.

Response: I looked over the reference carefully, and deleted the inaccurate reference [16].

>12) Lines 64 - 66: However, most of the cultivated soil types in the central Zhejiang basin are cohesive soil, which is easy depositing and compacting, and the penetration resistance is relatively high [17]. This reference [17] about rotary tillage in Malaysia, please revise.

Response: I looked over the reference carefully. By quoting references, author means that cohesive soil is easy to deposit. The explanation was as follows (line 67-69):

However, most of the cultivated soil types in the central Zhejiang basin are cohesive soil. Cohesive soil is easily deposited and compacted, and the penetration resistance is relatively high [16].

>13) Lines 71 - 73: (Solid plough bottom is formed under a long-term maintenance of rotary tillage, which makes a negative impact on root growth and nutrition acquisition, further influences the yield of crops [3,19–23]). Reference [3] about crop rotation and did not mention rotary tillage, [19] studied Soil compaction–N interactions in barley, and the plants were grown in plastic cylinders no mentioning rotary tillage, reference [20] mentioned (Increasing size of agricultural implements……etc.) not just rotary tillage, reference [21]: (The repeated plowing in conventional tillage commonly results in the formation of dense plow pans that may seriously affect root growth) not rotary, also revise reference [22]. Just reference [23] including what you mentioned above. Please revise your references. Also, you need to edit your English writing ([to deposit] instead of: [depositing[; [, and further influencing] instead of: [, further influences]

Response: I looked over the reference carefully, and deleted the inaccurate references [3], [19-22].

>14) Lines 75 - 76: (In order to achieve the goal of combining agricultural machinery and agronomy, and at the same time, to meet the needs of agricultural sustainable development [25]). It seems that something is missing here, or maybe this is a completion of the previous sentences? If so, why do you separate the two sentences? Please revise.

Response: I looked over the paper carefully, and changed the reference [24] insertion position. The revises was as follows (line 74-78):

Therefore, according to different soil conditions and restrictive factors, choosing appropriate tillage practices is an important research direction in order to achieve the goal of combining agricultural machinery and agronomy, and meet the needs of agricultural sustainable development [19,20].

Methodology

>15) Show the technical specification of the tillage machines and tools. What is the width of tillage tools cm?

Response: I showed the technical specification of the tillage machines and tools. The explanation was as follows (line 110-111):

The tillage width of RT, VRT, and VSRT were 1.8 m, 1.2 m, and 1.2 m. The designed tillage depth of RT, VRT and VSRT were 20 cm, 25cm, and 40 cm.

>16) Ignoring the working depth, show the technical differences between VRT and VSRT machines.

Response: I showed the technical differences between VRT and VSRT machines. The explanation was as follows (line 113-115):

Except tillage depth, there were differences between VRT and VSRT machines. The spiral blade of VSRT machine provided better soil smashing capacity and softer soil than the transverse blade of VRT machine.

>17) Why you did not follow the same time nodes for both years? This is is to avoid the effect of the different climates, which you confirmed in lines 359 to 361: (On the other hand, in 2020, the late growth period of broccoli was affected by diseases, which impeded the growth of flower balls, 360 resulting in the decline of broccoli yield).

Response: I added the explanation of the time nodes for both years. The explanation was as follows (line 101-102):

The start time of the test was adjusted according to the climate of the year, and 2021 is earlier than 2020.

>18) Line 121-122: (In 2020, the experiment plot was 60 × 20m which is divided into 15 experimental areas) what is the area for each plot? Is it 20 m × 4 m? And the same question for Line 122-123: (In 2021, the experiment plot was 65 × 15m which is divided into 15 experimental areas). Is it 20 m × 4.33 m? Show the area of each plot.

Response: The area of the experiment area after division is described in brackets. The explanation was as follows (line 126-129):

In 2020, the experimental plot was 60 × 20 m, which was further divided into 15 ex-perimental areas (12 × 6.6 m). In 2021, the experimental plot was 65 × 15 m, which was further divided into 15 experimental areas (13 × 5 m).

>19) You should show the diameter and the height of the auger cylinder that was used for collecting bulk density data.

Response: I show the diameter and the height of the auger cylinder. The explanation was as follows (line 153-155):

A fixed volume (Φ50.46×50 mm, 100 cm3) soil auger (Nanjing Soil Instrument Factory Co., Ltd, Nanjing, China) was used to collect soil samples, and samples were taken from the 0—40 cm soil layer.

>20) You should show how many samples you took per plot for bulk density and soil penetration resistance, and when did you take the samples before and after tillage, because you have two tillage in every experiment (Plough cultivated land two times with rotary tillage machine → Harrowing ground using the flat harrow linked to rotary tillage machine). So show when did you collect the data? Before rotary and after harrowing? Or immediately before planting?

>21) Show clearly that you collected soil samples for bulk density and water content with a 10 cm soil sampling interval depth.

Response: I looked over the paper carefully, and the explanation was as follows (line 153-156):

A fixed volume (Φ50.46×50 mm, 100 cm3) soil auger (Nanjing Soil Instrument Factory Co., Ltd, Nanjing, China) was used to collect soil samples, and samples were taken from the 0—40 cm soil layer. The samples were taken with a 10 cm interval depth. 5 samples were taken from each experimental area before planting.

>22) You have collected the data before and after plowing, it would be very clear to show the effect of tillage on soil physical properties by comparing physical properties before and after tillage.

Response: I looked over the paper and thought carefully. I think this paper mainly focuses on the differences of soil properties after different tillage methods, rather than the differences of soil properties before and after tillage, so it is not explained in details.

>23) How did you determine the soil moisture content? You should show the procedure and the equation.

Response: The soil moisture content test procedure and the equation was as follows (line 157-160):

The equation for calculating SWC was as follows:

where Wf and Wd represent fresh weight and dry weight of soil.

>24) Your calculation method of the bulk density is confusing, what do you mean by the weight of the soil auger? Did you put the soil auger in the oven? What do you mean by fresh soil? Is it the soil before drying? I'm confused I do not know what you mean. You should weigh the soil sample before putting it in the oven and after that. Use the dry soil mass to calculate the bulk density, and the weight difference between wet and dry soil to calculate the moisture content. This is not written here. I do not know from where you got the equation that you used to calculate the bulk density. The bulk density is calculated as (oven-dried soil mass/soil volume). You mentioned reference (26), but this reference did not mention this equation. This reference (Methods of Soil Analysis: Part 1 Physical and Mineralogical Methods, 5.1, Second Edition) stipulated on page 366 that: (Transfer the soil to a container, place it in an oven at 105°C until constant weight is reached, and weigh it. The bulk density is the oven-dry mass of the sample divided by the sample volume). You need to make a major explanation and revision here.

Response: I looked over the reference carefully, and modified inaccurate descriptions. The revises was as follows (line 161-163):

Transfer the soil to a container, place it in an oven at 105°C until constant weight is reached, and weigh it. The bulk density is the oven-dry mass of the sample divided by the sample volume (100 cm3).

>25) Lines 158 and 159: (After tillage practice, the soil penetration resistance was measured by the soil penetration resistance tester) is this tester a soil penetrometer? If so, what is the cone angle of the penetrometer (o) and the conical point (cm2) of the penetrometer that you used?

>26) Show a picture of the soil penetration resistance tester (Nanjing Soil Instrument Factory Co., Ltd).

Response: I added the explanation of the tester and the picture of it. The revises was as follows (line 165-170):

After tillage, the soil penetration resistance was measured by the portable electronic soil penetration resistance tester (Nanjing Soil Instrument Factory Co., Ltd, Nanjing, China). Figure 2 shows the tester and the test process.

Figure 2. Portable electronic soil penetration resistance tester

>27) Line 174-176: Root length, how many samples per plot did you take, and how did you choose it?

Response: I looked over the paper carefully, and added the description of sampling process. The revises was as follows (line 184-186):

5 broccoli samples were taken from each experimental area to measure the root length. First determine the midpoint of the diagonal as the central sampling point, and then select four points on the diagonal with the same distance from the central sampling point.

>28) Line 178-179: Broccoli characters & yield. How many samples per plot did you take and how did you choose them?

Response: I added the description of sampling process. The revises was as follows (line 190):

Broccoli characteristics and yield were measured with the root length at the same time.

Result

>29) In your figures (figure 5 and figure 6) put every two similar factors together in a horizontal line, this would make it easy to compare.

Response: I have put every two similar factors together in a horizontal line.

>30) Do you believe that deep plowing will always contribute to an increase in the broccoli yield, noting the comparison between RT and VRT, and do you see that your results are definitive and decisive? Maybe the deeper is most effective.

Response: As mentioned above, in a certain range of depth, increasing tillage depth and soil fragmentation will contribute to an increase in the broccoli yield. This paper does not explore the impact of deeper tillage depth (deeper than VSRT) on the broccoli yield. The reason is that, generally, too much tillage depth is not required before broccoli planting, and the deeper the tillage depth, the greater the power consumed by the machinery.

>31) Please show the differences % (the increment or decrement ratio) between each soil property before and after tillage for RT, VRT, and VSRT. So readers could see the effect ratio.

Response: I added the differences % (the increment or decrement ratio) between each soil property. The revises was as follows (line 206-212, line 223-225, line 234-238):

The biggest difference was observed in the 20-30 cm soil layer, the SWC under VSRT was 1.6% and 1.9% more than RT and VRT respectively. The opposite phenomenon was observed in the 0—10 cm soil layer, in which the SWC of VSRT was less than that of RT and VRT. In 2021, there was a significant difference of the SWC under VSRT in the 0—20 cm soil layer, and there was little difference at 30—40 cm. The biggest difference was observed in the 10-20 cm soil layer, the SWC under VSRT was 0.9% and 1.3% more than RT and VRT respectively.

The biggest difference was observed in the 30-40 cm soil layer in 2020, the soil pene-tration resistance under VSRT was 52.6% and 41.8% less than RT and VRT respectively.

The biggest difference was observed in the 20-30 cm soil layer, the SBD under VSRT was 7.4% and 8.8% less than RT and VRT respectively. In 2021, especially, the SBD of the 10—40 cm soil layer showed a significant decline under VSRT. The biggest difference was observed in the 20-30 cm soil layer, the SBD under VSRT was 14.6% and 15.1% less than RT and VRT respectively.

Discussion

>32) Although VRT is deeper and the tillage tool is lighter than RT, your result showed that RT in the top layer of soil has a lower bulk density than VRT in 10 to 30 cm as in Figure 4, please explain and justify.

Response: I looked over the paper carefully, and the revises was as follows (line 323-326):

The SBD under VRT is more than RT. VRT has a high-speed rotating transverse blade, so the soil after cultivation has a high degree of fragmentation. The highly crushed soil is easy to deposit naturally with the influence of rain, and the SBD is high after deposition.

>33) What was the effect of root length and DMA on broccoli yield, with notice that root length and DMA under VRT were mostly higher than that under RT? And the yield under RT was higher than under VRT. You should explain and discuss this.

Response: I looked over the paper carefully, and the revises was as follows (line 372-373):

VRT had higher soil fragmentation, reduced the resistance of broccoli root growth, so the root length of broccoli was longer than RT.

>34) The Soil penetration resistance under RT was higher than that under VRT in all soil layers during both years, and plant DMA, DMA accumulation, and DMA rate of Broccoli plant floret ball, florets ball diameter (cm), and the broccoli yield under RT were higher than that under VRT, is the effect of penetration resistance is ineffective or what? Discuss and explain.

Response: I looked over the paper carefully, and the explanation was as follows (line 393-395):

Compared with VRT, RT increased the yield of broccoli. This effect was mainly because VRT had higher soil fragmentation and higher SBD, which inhibited root respiration of broccoli, thus reducing broccoli DMA.

>35) Lines 297 and 298 (so the soil after tillage is looser), the right is to say (so the soil after tillage is loosened).

Response: I have revised this error.

>36) Lines 316 and 317: (Compared with 2020, there is more rainfall in each month, resulting in higher SWC and lower SPR in 2021. However, long-term wet soil leads to more dense soil, so there is higher SBD in 2021). You mentioned that bulk density in 2021 was higher but in line 95 you mentioned bulk density in 2020 was 1.47 g/cm3 and in line 98 you mentioned bulk density in 2021 was 1.42 g/cm3, explain. Is the effect of higher water content on bulk density positive or negative? What is the relation between soil penetration resistance, bulk density, and soil water content?

Response: I looked over the paper carefully, and the explanation was as follows (line 396-398):

The soil penetration resistance under RT was high, but the overall soil penetration resistance was low, the effect of soil penetration resistance on broccoli growth was smaller than that of SBD and SWC.

>37) Lines 316 and 317: (Compared with 2020, there is more rainfall in each month, resulting in higher SWC and lower SPR in 2021). But when we look at Figure 2 we see that the SWC in 2020 in the 10 to 40 cm layer was higher than that of 2012, explain.

Response: The SWC in 2021 in the 0-20 cm layer was higher than that of 2020. I have revised this error, and the explanation was as follows (line 351-353):

Compared with 2020, there was more rainfall in each month, resulting in higher SWC and lower soil penetration resistance in 2021 in 0-20 cm soil layer.

>38) Lines 361 to 363 (Therefore, the results of this study show that VSRT, as a new deep tillage method, can be used to improve the adverse effects of long-term rotary tillage on soil properties and increase the yield of broccoli [44,45]). When returning to reference [44] which is: (de Pascale, S.; Maggio, A.; Barbieri, G. Soil Salinization Affects Growth, Yield and Mineral Composition of Cauliflower and Broccoli. European Journal of Agronomy 2005, 23, 254–264, doi:10.1016/j.eja.2004.11.007), and reference [45] which is (Wurr, D.C.E.; Hambidge, A.J.; Fellows, J.R.; Lynn, J.R.; Pink, D.A.C. The Influence of Water Stress during Crop Growth on the Postharvest Quality of Broccoli. Postharvest Biology and Technology 2002, 25, 193–198, doi:https://doi.org/10.1016/S0925-5214 (01) 00171-5) there is no any mentioning of VSRT, the first paper about planting stress (water and heat stress) and the second one about soil salinity. I do not know from where you got that. When you cite a reference you should be sure that what you referred to this reference is exist in it.

Response: I looked over the references carefully, and deleted the wrong references.

>39) You should be sure about your references that they include the information you referred to them, please check all your references.

Response: I looked over all of the references carefully, deleted the wrong references, and renumbered the references.

Reviewer 2 Report

The work is interesting, as it deals with deep tillage and its impacts on the soil and broccoli performance. We can witness a hotspot of managing the soil in depth layer, one unique aspect of soil management in intensifying its eco-servicing functionalities. The vertical smashing rotary tillage provided as deep as 40 cm soil loosening, which largely enhanced soil functionalities. Observed improvements of soil properties, root length and biomass, and up-ground biomass output of broccoli revealed its effectiveness of managing the soil condensed or hardened due to the local soil type and management practices. The acquired results add to the archive of soil science and expands our knowledge.

Problems in general:

Figure 1 lacks scales and thus fails to illustrate the size of each machines and related parts.

In the materials and method section: It is necessary to explain what the power is required for each machine. Also, the field working efficiency should be provided. These and other indicators should be provided in detail.

English text:

The text is very poorly written. Major revision is required.

Author Response

Response to Reviewers' comments

>Reviewer #2:

The work is interesting, as it deals with deep tillage and its impacts on the soil and broccoli performance. We can witness a hotspot of managing the soil in depth layer, one unique aspect of soil management in intensifying its eco-servicing functionalities. The vertical smashing rotary tillage provided as deep as 40 cm soil loosening, which largely enhanced soil functionalities. Observed improvements of soil properties, root length and biomass, and up-ground biomass output of broccoli revealed its effectiveness of managing the soil condensed or hardened due to the local soil type and management practices. The acquired results add to the archive of soil science and expands our knowledge.

>1) Figure 1 lacks scales and thus fails to illustrate the size of each machines and related parts.

Response: I added a brief introduction about the operating dimensions of machinery to illustrate the dimensions of the machine, and the explanation was as follows (line 110-111):

The tillage width of RT, VRT, and VSRT were 1.8 m, 1.2 m, and 1.2 m. The designed tillage depth of RT, VRT and VSRT were 20 cm, 25cm, and 40 cm.

>2) In the materials and method section: It is necessary to explain what the power is required for each machine. Also, the field working efficiency should be provided. These and other indicators should be provided in detail.

Response: All the machines power is provided by the tractor, and the explanation was as follows (line 111-113):

The operation power of the three tillage machines was provided by the tractor (Kubota M954KQ, 70.8kW).

>3) The text is very poorly written. Major revision is required.

Response: The manuscript was already corrected by a native English speaker.

Round 2

Reviewer 1 Report

It is OK, you responded well to the revision.

Author Response

Dear reviewers:

    Thank you for your comments on our manuscript ‘Effects of vertical smashing rotary tillage on root growth characteristics and yield of broccoli’, which really helps to improve our manuscript quality.

    Thanks again!

Reviewer 2 Report

The manuscript still requires extensive revision, primarily due to the non-standardized text writing.

Author Response

    The manuscript still requires extensive revision, primarily due to the non-standardized text writing.

    Response: According to the reviewers' suggestions, the manuscript was revised and the manuscript was already corrected by a native English speaker again (revised parts were marked in red).
